# Hybrid Student-Teacher Large Language Model Refinement for Cancer Toxicity Symptom Extraction

Reza Khanmohammadi[1,*], Ahmed I. Ghanem[2,3], Kyle Verdecchia[2], Ryan D. Hall[2],
Mohamed Elshaikh[2], Bing Luo[2], Benjamin Movsas[2], Hassan Bagher-Ebadian[2,5,6,7], Indrin J. Chetty[4],
Tuka Alhanai[8], Kundan Thind[2,9], Mohammad M. Ghassemi[1]

[1]Department of Computer Science and Engineering, Michigan State University, East Lansing, MI, USA
[2]Department of Radiation Oncology, Henry Ford Cancer Institute, Detroit, MI, USA
[3]Alexandria Department of Clinical Oncology, Alexandria University, Egypt
[4]Department of Radiation Oncology, Cedars Sinai Medical Center, Los Angeles, CA, USA
[5]Departments of Radiology and Osteopathic Medicine, Michigan State University, East Lansing, MI, USA
[6]Department of Physics, Oakland University, Rochester, MI, USA
[7]Department of Oncology, Wayne State University, Detroit, MI, USA
[8]Department of Computer Engineering, New York University Abu Dhabi, Abu Dhabi, UAE
[9]Department of Medicine, Michigan State University, East Lansing, MI, USA
[*]Corresponding author: `khanreza@msu.edu`

*Abstract*—**Large Language Models (LLMs) offer significant potential for clinical symptom extraction, but their deployment in healthcare settings is constrained by privacy concerns, computational limitations, and operational costs. This study investigates the optimization of compact LLMs for cancer toxicity symptom extraction using a novel iterative refinement approach. We employ a student-teacher architecture, where the teacher model, GPT-4o, dynamically selects the most effective strategy for the student models (Zephyr-7b-beta and Phi3-mini-128) between prompt refinement, Retrieval-Augmented Generation (RAG), and fine-tuning. Our experiments on 294 clinical notes covering 12 post-radiotherapy toxicity symptoms demonstrate the effectiveness of this approach. Using 5-fold cross-validation, we observed significant improvements in F1 scores across all symptoms. The Phi3 model showed an average F1 score increase of 26%, while Zephyr achieved a 13% improvement. Notably, these enhancements were achieved at substantially lower costs, with Phi-3 being 48 times cheaper and Zephyr 30 times cheaper than GPT-4o. These results highlight the potential of iterative refinement techniques to enhance the capabilities of compact LLMs for clinical applications, offering a balance between performance, cost-effectiveness, and privacy preservation in healthcare settings.**

*Index Terms*—**Prompt Refinement, Fine-tuning, Large Language Models, Toxicity Symptom Extraction**

## I. INTRODUCTION

### A. Optimized LLMs in Clinical Settings

**Opportunities and Challenges:** Large Language Models (LLMs) have significant potential in clinical informatics by processing the growing volume of unstructured data from electronic health records (EHRs), clinical notes, and patient reports [1]. Advanced natural language processing (NLP) tools are crucial for extracting valuable information, such as identifying symptoms, to improve patient care and support clinical decision-making [2]. However, integrating LLMs into healthcare is constrained by privacy concerns, computational demands, and cost, as described in the following sections.

**Privacy Concerns and Resource Limitations:** A primary concern for healthcare institutions is ensuring patient privacy [3], which often necessitates deploying LLMs on-premises to avoid transmitting sensitive data to external servers. However, running large LLMs locally requires significant computational resources that many facilities cannot easily support. Additionally, using advanced third-party models like GPT-4, while powerful, is often financially unsustainable due to high usage costs [4]. These constraints underscore the need for cost-effective, locally deployable solutions that balance performance and resource demands.

**Smaller LLMs and Their Limitations:** To address privacy concerns, computational resource limitations, and cost constraints, healthcare institutions are exploring smaller LLMs better suited for local deployment [5]. These models, with fewer parameters than larger ones like GPT-4, help reduce strain on resources and costs. However, they often show reduced performance. For instance, while GPT-4 achieves 87% and 86% accuracy on medical benchmarks like MedMCQA and PubMedQA, smaller models typically score 20-30% lower [6]. This performance gap highlights the challenges smaller models face in tasks like medical QA and clinical reasoning.

**Optimization Challenges:** The main challenge is optimizing smaller LLMs to accurately process clinical data despite resource constraints. This is particularly important for tasks like symptom extraction and interpretation, which are critical in reducing medical errors. Medical errors cause 200,000 preventable deaths and harm 400,000 patients annually in the U.S., costing the healthcare system $20 billion [7]. Improving

the performance of smaller LLMs can help enhance patient outcomes, lower costs, and ensure data privacy is maintained.

### B. Clinical Symptom Extraction with LLMs

Recent studies have explored LLMs for extracting clinical information from unstructured EHR texts. Mahbub *et al.* [8] used zero-shot learning with Flan-T5 for SUD severity extraction, outperforming rule-based approaches. Reese *et al.* [9] found GPT-4's performance in clinical diagnostics to be sensitive to prompt formulation. Shyr *et al.* [10] demonstrated GPT-3.5 Turbo's efficacy in zero- and few-shot settings for rare disease phenotype extraction. Guevara *et al.* [11] showed fine-tuned Flan-T5 models' superiority in extracting social determinants of health, especially with synthetic data augmentation. These studies underscore LLMs' potential to enhance critical health information extraction from clinical texts, improving symptom and phenotype identification for effective radiation oncology toxicity management.

### C. Iterative LLM Refinement Techniques

**Single LLM Refinement:** Recent advancements in LLMs have focused on iterative refinement techniques to enhance performance across diverse tasks. Chen *et al.* [12] demonstrated that multiple refinement rounds led to small but consistent improvements, with COMETDA increasing from 0.8427 to 0.8478 and COMETQE from 0.1083 to 0.1153. Human evaluations showed that 32% of evaluators preferred the refined German-to-English translations, and 31% preferred the refined Chinese-to-English translations, indicating improved fluency and naturalness. Building on this concept, Xiong *et al.* [13] developed the IPR framework, which outperformed established baselines in complex interactive tasks. Madaan *et al.* [14] further advanced this approach with *Self-Refine*, yielding significant improvements of 32% in Sentiment Reversal, 49% in Dialogue Response, and 30% in Constrained Generation, without additional training data. Yan *et al.* [15] extended these principles to enable resource-efficient performance enhancements in less capable models. n the domain of bioinformatics, Chen *et al.* [16] applied iterative prompt refinement to substantially improve ChatGPT's accuracy in extracting gene relationships and biological pathways, increasing coverage by about 11% in pathway reconstruction.

**Collaborative LLM Refinement:** Recent advancements have explored collaborative frameworks utilizing multiple LLMs. Lee *et al.* [17] introduced LLM2LLM, where a teacher LLM augments small datasets to enhance student LLM performance in low-data scenarios. Zhang *et al.* [18] developed TS-Align, aligning LLMs with human preferences without manual annotations. Saha *et al.* [19] demonstrated improvements in student LLM performance through teacher-generated personalized explanations. Yuan *et al.* [20] showed that teacher-generated analogies substantially improve student LLMs' scientific concept comprehension and question-answering capabilities. These studies collectively demonstrate the efficacy of collaborative LLM frameworks in enhancing performance

across diverse domains and tasks, particularly in scenarios with limited data or complex reasoning requirements.

### D. LLM Refinement in Symptom Extraction

Khanmohammadi *et al.* [21] introduce a student-teacher architecture for prostate cancer radiotherapy symptom extraction, where the large-scale Mixtral model serves as the student and GPT-4 as the teacher. Their approach focuses specifically on iterative prompt refinement; the teacher model repeatedly refines the prompts given to the student to improve performance. While this method resulted in a significant improvement, raising the average F1 score from 0.49 to 0.73, its focus was on refining a single, large student model using one primary technique. This foundational work establishes the viability of the student-teacher paradigm for this task, demonstrating that iterative prompt refinement can significantly enhance performance in a specialized clinical domain.

### E. Remaining Gaps

While recent studies have advanced LLM integration in clinical symptom extraction, several areas require further investigation:

- Application of student-teacher frameworks to more compact LLMs ($<$10 billion parameters) for on-premises clinical deployment.
- Integrating RAG into the student-teacher framework for improved clinical prompt optimization.
- Optimization of fine-tuning strategies for domain-specific adaptations in clinical use.
- Expansion of the teacher model's role in dynamically guiding student learning process.

These directions address challenges in deploying efficient, accurate, and privacy-preserving LLMs in clinical settings.

### F. Contributions of This Work

This study investigates several aspects of the student-teacher framework for clinical symptom extraction:

- Application of the framework to smaller LLMs (7 and 3.8 billion parameters) for clinical settings.
- Integration of RAG within prompt refinement to enhance contextual understanding and accuracy.
- Exploration of iterative fine-tuning for domain adaptations in clinical contexts.
- Implementation of an advanced student-teacher framework where the teacher model optimizes refinement strategies as a decision agent.

Project details, including full prompt templates, are available on GitHub: https://github.com/Ledengary/hybrid-llm-refinement.

## II. EXPERIMENTS

### A. Data Description

Our dataset comprises $D = \{x_1, x_2, \ldots, x_{294}\}$ clinical notes from 100 unique prostate cancer patients treated with 78 Gy radiotherapy (RT) between 2013 and 2020, documented beyond 6 months post-RT to focus on long-term toxicities.

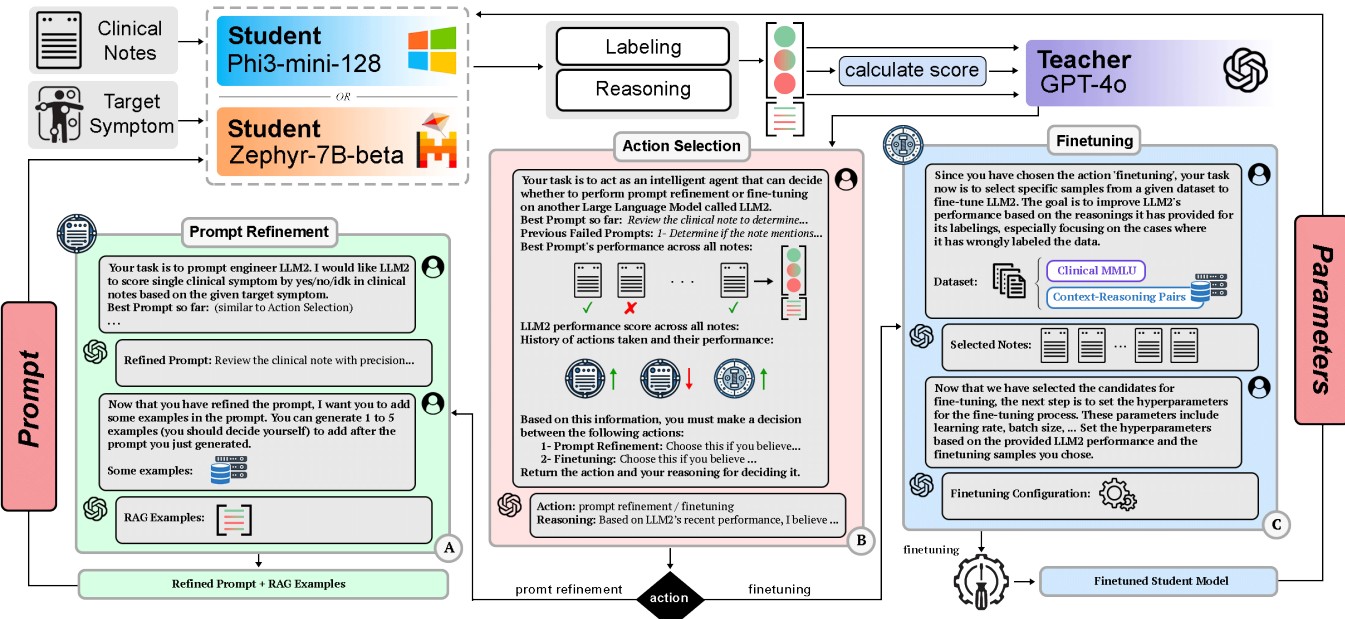

Fig. 1. The diagram illustrates the iterative refinement method, involving a student model (Phi3-mini-128 or Zephyr-7B-beta) and a teacher model (GPT-4o). The process starts with the student model receiving clinical notes and a target symptom, generating initial labels and reasoning. The teacher model then assesses performance and decides (B) between prompt refinement (A) and fine-tuning (C). The teacher model enhances prompts with RAG examples, selects samples and hyperparameters for fine-tuning, and dynamically chooses between these methods in a hybrid approach. This process is applied iteratively to improve the student model's symptom extraction capabilities.

Patients contributed multiple notes over time. To avoid leakage in 5-fold cross-validation, we enforced patient-level splits: all notes from a patient (grouped by their surrogate Medical Record Number) were assigned entirely to either training or test folds. Each note $x_i$ is labeled with $y_i \in \{-1, 0, 1\}$, where $y_i = -1$ indicates absence, $y_i = 0$ unknown status, and $y_i = 1$ presence of one of twelve common post-RT toxicities ($S$): Cystitis, Dysuria, Erectile Dysfunction, Hematuria, Incontinence, Nocturia, Proctitis, Rectal Bleeding, Stricture, Urgency, Urinary Obstruction, and Urothelial Carcinoma. This explicit "unknown" ($y_i = 0$) label allows the model to abstain on ambiguous or conflicting notes rather than forcing a decision. To ensure patient privacy, we applied the *Stanford & Penn MIDRC Deidentifier* [22] to automatically remove all protected health information (PHI), replacing it with realistic surrogates while maintaining clinical context.

To ensure robust evaluation, we employed stratified 5-fold cross-validation, partitioning $D$ into 5 disjoint subsets $D_1, D_2, \ldots, D_5$, ensuring each subset maintained the same symptom distribution as in the overall dataset. For each fold $k$, the training set is defined as $D_{\text{train}}^{(k)} = D \setminus D_k$ and the test set as $D_{\text{test}}^{(k)} = D_k$, where $k = 1, 2, \ldots, 5$, resulting in 236 notes for training and 58 notes for testing.

In addition to the clinical notes, we utilized selected categories from the Massive Multitask Language Understanding (MMLU) dataset [23] to form a clinical subset MM for fine-tuning. The categories — anatomy, clinical knowledge, college medicine, human sexuality, medical genetics, and professional medicine — yielded 1225 records. Based on the Superficial Alignment Hypothesis [24], which suggests that a model's pre-

trained knowledge can be refined with a relatively small set of examples, this subset was used to enhance the model's clinical knowledge for symptom extraction.

### B. Data Preprocessing

Our preprocessing, designed to support the RAG component, involves embedding each clinical note $x_i$ into a 768-dimensional vector $z_i$ using `Bio_ClinicalBERT`:

$$z_i = f_{\text{BERT}}(x_i) \in \mathbb{R}^{768} \tag{1}$$

In addition, GPT-4o generates context-reasoning pairs $C_R(x_i) = (c_i, r_i)$ for each note, where $c_i$ represents relevant context and $r_i$ represents reasoning for the label $y_i$. These embeddings and $C_R$ pairs are stored in a vector database, creating a semantically rich dataset. This dataset serves as the input for the iterative refinement process, allowing for context-aware improvements in the student model's performance within the RAG framework.

### C. Iterative Refinement

*1) Concepts and Terminologies:* The iterative refinement process in a student-teacher framework has two main components:

**1. Student Model:** The student model, such as Phi3-mini-128 or Zephyr-7B-beta, is responsible for the initial task of symptom extraction from clinical notes. Formally, let $f_{\text{student}}$ represent the student model, which takes an input $z_i$ (the embedded clinical note) and a prompt $p$, and generates an output prediction $\hat{y}_i$:

$$\hat{y}_i = f_{\text{student}}(z_i, p) \tag{2}$$

where $z_i \in \mathbb{R}^{768}$ is the embedded vector of note $x_i$ and $\hat{y}_i \in \{-1, 0, 1\}$ is the predicted label.

**2. Teacher Model:** The teacher model, $f_{\text{teacher}}$, GPT-4o in our case, oversees the refinement process. It evaluates the student model's performance using a performance metric such as the macro F1-score $s = \text{F1-macro}(Y, \hat{Y})$, where $Y$ are the true labels and $\hat{Y}$ are the predicted labels from the student model. The teacher iteratively refines the prompts or fine-tunes the student model's parameters to improve this performance score.

*2) Iterative Refinement Process:* The iterative refinement process is as follows:

**I. Initial Classification:** The student model first processes the notes for a specific symptom $(S)$, using a structured prompt to classify the presence of the symptom. At iteration $t = 0$, the student model generates an initial prediction $\hat{y}_i^0$ for each note $x_i$ using an initial prompt $p_0$:

$$\hat{y}_i^0 = f_{\text{student}}(z_i, p_0) \tag{3}$$

The student model classifies each note as *yes*, *no*, or *idk* and provides its reasoning. This initial classification forms the baseline for further refinements in subsequent iterations.

**II. Performance Evaluation:** The student model's outputs are evaluated by comparing the predicted labels $\hat{Y}_t = \{\hat{y}_i^t\}$ with the ground truth $Y$. We denote $s_t$ as the performance score at iteration $t$, calculated using the defined macro F1-score.

**III. Refinement by the Teacher:** Based on the performance score $s_t$ and the reasoning provided by the student model, the teacher model $f_{\text{teacher}}$ decides whether to refine the prompt or fine-tune the model parameters. The teacher evaluates the student's current performance and action history to determine the next refinement step.

**IV. Action Selection:** The teacher model $f_{\text{teacher}}$ chooses between two refinement strategies: $d_t = $ Prompt Refinement if prompt refinement is deemed more effective, and $d_t = $ Fine-Tuning otherwise. In prompt refinement, the teacher generates a refined prompt $p_{t+1}$, while in fine-tuning, it adjusts the student model parameters $\theta$ by minimizing the loss $L(Y, \hat{Y}_t)$ using gradient descent with learning rate $\eta$:

$$\theta_{t+1} = \theta_t - \eta \nabla_{\theta_t} L(Y, \hat{Y}_t) \tag{4}$$

**V. Iteration and Application:** The selected refinement strategy is applied, and the student model processes the data again in the next iteration. If performance improves, a new round begins. The process continues for 16 rounds per epoch, with each iteration $t$ yielding a new prompt $p_t$ or updated parameters $\theta_t$.

**VI. Epoch Progression:** Each epoch consists of 16 refinement rounds. After each epoch, the best-performing prompt $p_{\text{best}}$ and the corresponding model parameters $\theta_{\text{best}}$ are selected. The process continues until no further improvement in $s_t$ is observed, indicating convergence. The final output is the optimized student model $f_{\text{student}}$ with the best-performing configuration.

*D. Methods Investigated*

*1) Prompt Refinement Approach:* The prompt refinement approach is a key component of our iterative refinement process, focusing on optimizing the prompts used by the student models for symptom extraction. This approach consists of two main steps:

**I. Prompt Refinement:** When this action is selected, the teacher model $f_{\text{teacher}}$ (GPT-4o) refines the prompt $p_t$ used by the student model. The refinement is based on the student model's current performance score $s_t$. The goal is to generate an improved prompt $p_{t+1}$ that better guides the student model $f_{\text{student}}$ in extracting symptoms from clinical notes:

$$p_{t+1} = f_{\text{teacher}}(s_t, p_t) \tag{5}$$

The teacher model analyzes the student's performance and generates a refined prompt aimed at improving the extraction capabilities of the student model $f_{\text{student}}(z_i, p_{t+1})$.

**II. RAG Example Generation:** After refining the prompt, the next step involves generating RAG examples. For each clinical note $x_i$, the system queries a vector database to retrieve semantically similar context-reasoning pairs $C_R(x_i)$. These pairs are represented as:

$$C_R(x_i) = \{(c_{i_1}, r_{i_1}), (c_{i_2}, r_{i_2}), (c_{i_3}, r_{i_3})\} \tag{6}$$

where $c_j$ denotes the context, and $r_j$ represents the reasoning associated with each retrieved note $x_j$. The teacher model utilizes these pairs to create examples that are used to further refine the student model's understanding. Next, the teacher model $f_{\text{teacher}}$ uses the refined prompt $p_{t+1}$ and the retrieved $C_R(x_i)$ pairs to generate between one and five (n) RAG examples $E$, which enhance the student model's understanding:

$$E = f_{\text{teacher}}(p_{t+1}, C_R(x_i)) = \{e_1, \ldots, e_n\} \tag{7}$$

The refined prompt $p_{t+1}$ is updated by appending the generated RAG examples $E$, i.e., $p_{t+1} = p_{t+1} + E$, and used by the student model in the next iteration of symptom extraction.

*2) Fine-Tuning Approach:* The fine-tuning approach adapts the student model for improved clinical symptom extraction, consisting of these steps:

**I. Sample Selection:** The teacher model $f_{\text{teacher}}$ strategically selects samples from both the clinical MMLU dataset $M$ and the context-reasoning pairs $C_R$ derived from our clinical notes. The sample selection is guided by the student model's performance on previous iterations, with a focus on areas where the model has shown weakness or failure. The selected samples directly address the student model's shortcomings in symptom identification and classification.

**II. Fine-Tuning Configuration:** Once the relevant samples have been selected, the teacher model determines the optimal hyperparameters for fine-tuning the student model $f_{\text{student}}$. This involves adjusting key parameters based on the architecture and the target modules used in parameter-efficient fine-tuning (PEFT), such as attention heads and transformer layers.

By using these tailored samples and fine-tuned hyperparameters, the student model undergoes targeted fine-tuning, ensuring that the process addresses the identified weaknesses.

This approach leads to an efficient and effective improvement in the model's ability to accurately extract and classify symptoms from clinical notes.

*3) Hybrid Approach:* The hybrid approach combines the strengths of prompt refinement and fine-tuning, enabling dynamic adaptation based on the model's performance and needs. Here, the teacher model acts as an intelligent agent, deciding between prompt refinement and fine-tuning at each iteration. The process is outlined as follows:

**I. Action Selection:** $f_{\text{teacher}}$ is presented with a comprehensive prompt that includes: (a) the best-performing prompt so far, (b) previous prompts that were less effective, (c) the student model's performance across all notes, including ground truth labels, output labels, and reasoning, and (d) a history of previous actions taken and their resulting performance metrics. Based on this information, the teacher model selects an action, denoted as $a_t \in \{\text{Prompt Refinement}, \text{Fine-Tuning}\}$, representing the refinement strategy for the next iteration. The teacher model returns its decision as a JSON object, including the chosen action $a_t$ and a brief explanation for why it believes this action is the most effective next step.

**II. Action Execution:** Based on the teacher's decision $a_t$: (a) if prompt refinement is chosen, the process proceeds as described in Section II-D1, and (b) if fine-tuning is chosen, the process proceeds as described in Section II-D2. To effectively consider the history of previous interactions and actions taken, we embedded this information directly into the prompt provided to the teacher model. This embedded history includes essential details of past refinements, such as the specific actions taken (prompt refinement or fine-tuning), the resulting performance metrics $s_t$, and the hyperparameters used in previous iterations. By incorporating this historical context within the prompt, we enable the teacher model to make informed decisions based on prior outcomes without the need for maintaining a costly and extensive chain of interactions. This approach allows the teacher model to adapt its strategy in subsequent rounds. For instance, if prompt refinement has been attempted multiple times without significant improvement, the teacher model might decide to switch to fine-tuning, or vice versa. This method effectively balances the need for historical context with the constraints of the model's context window and computational efficiency considerations.

### E. Student and Teacher Evaluation Metrics

In this study, we evaluate our student models using a stratified 5-fold cross-validation approach, measuring accuracy and F1-macro scores after each symptom annotation. Final performance metrics are reported as averages across all folds. We assess computational costs based on prompt length and model weights, calculating expenses for processing tokens with GPT-4o at $5.00 and $15.00 per million input and output tokens, respectively. Energy consumption is evaluated by tracking power usage during model inference, converted to monetary costs using the average U.S. electricity rate of 16.88 cents per kilowatt-hour. To balance performance and cost-effectiveness, we introduce a Performance-Cost Ratio (PCR), calculated by dividing the performance metric (accuracy or F1-score) by the associated cost. This approach, combining cross-validation with cost analysis and PCR, ensures a thorough understanding of both clinical efficacy and operational efficiency of the student models in realistic healthcare settings.

## III. RESULTS

**Performance Improvement Across Refinement Techniques:** During the iterative refinement process, both student models showed significant performance gains, with the choice of technique yielding distinct outcomes. For the Zephyr model, RAG refinement provided the largest improvement, increasing the average F1 score from 0.32 to 0.73, a mean increase of 0.41. The hybrid method also showed strong improvement, reaching a final score of 0.69 (a 0.38 mean increase), while fine-tuning alone yielded more modest gains, ending at 0.46.

The smaller Phi-3 model demonstrated even greater responsiveness to refinement. The RAG approach lifted its score from 0.40 ± 0.20 to 0.87 ± 0.09, the largest overall improvement with a mean increase of 0.46 and a standard deviation reduction of 0.11. The hybrid method was also highly effective, achieving a final score of 0.80 ± 0.14. In contrast, fine-tuning offered negligible performance gains for Phi-3, with a final score of just 0.42 ± 0.20.

**Test Set Performance and Cost Analysis:** Figure 2 shows Phi-3's performance across refinement techniques, reporting accuracy, F1-macro, and average cost per test note. The initial average accuracy score for Phi-3 was 0.446. After refinement, the Hybrid approach showed the highest improvement with an average accuracy of 0.636, closely followed by RAG at 0.629, while fine-tuning yielded minimal improvement to 0.512. For F1-macro scores, the initial performance was 0.307, with the Hybrid method achieving the highest refined average score of 0.567, followed by RAG at 0.509, while fine-tuning showed almost no improvement. GPT-4o, tested with both the initial Phi-3 prompt and the refined Hybrid prompt (selected as it performed best in both accuracy and F1-macro), demonstrated initial accuracy and F1-macro scores of 0.710 and 0.614 respectively, improving to 0.902 and 0.826 with the refined prompt. Notably, the average costs per test note varied significantly: GPT-4o was the most expensive at $7.23 \times 10^{-2}$, followed by fine-tuning at $3.7 \times 10^{-3}$, Hybrid at $1.5 \times 10^{-3}$, and RAG being the most cost-effective at $1.3 \times 10^{-3}$.

For the Zephyr model, the initial average accuracy was 0.364 across all methods. Post-refinement, the RAG approach demonstrated the highest improvement with an average accuracy of 0.593, followed by the Hybrid method at 0.573, while fine-tuning showed minimal improvement to 0.413. For F1-macro scores, the initial average performance was 0.315, with RAG achieving the highest refined average score of 0.487, followed by Hybrid at 0.442, and fine-tuning showing no substantial improvement at 0.345. GPT-4o, evaluated using both the initial Zephyr prompt and the refined RAG prompt (chosen for its superior performance in both accuracy and F1-macro), showed initial accuracy and F1-macro scores of 0.664 and 0.601 respectively, improving to 0.876 and 0.849

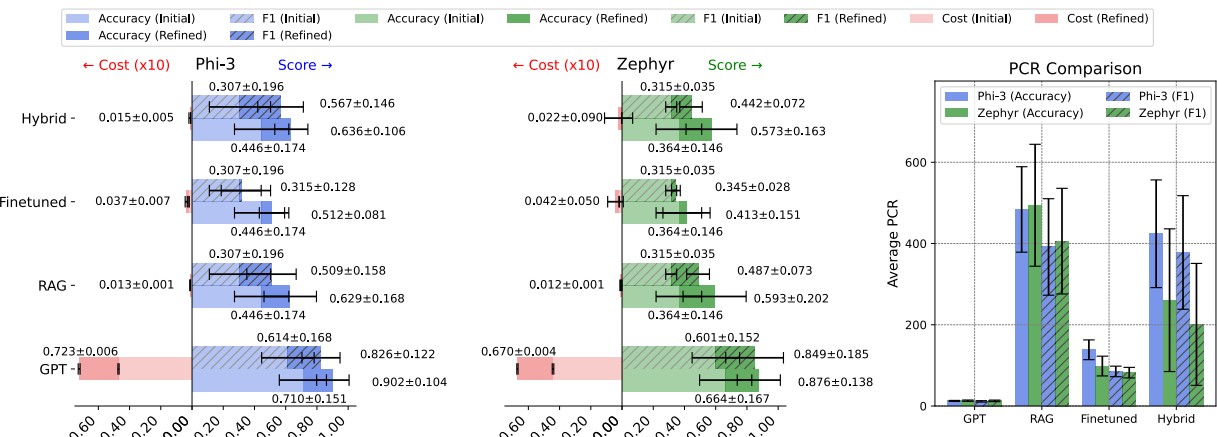

Fig. 2. Performance and cost analysis of Phi-3 and Zephyr models using 5-fold cross-validation. Left panels: Initial (brighter colors) and refined (darker colors) performance scores (blue and green bars) and associated costs (red bars) across different refinement techniques: Hybrid, Finetuned, RAG, and GPT-4o. Hatched bars represent F1-macro scores, smooth bars indicate accuracy. Values shown are averages across 12 toxicity symptoms, with each symptom's score being the mean of its 5-fold cross-validation results. Error bars represent standard deviations across symptoms. Right panel: Average Performance-Cost Ratio for refined Phi-3 and Zephyr models, illustrating the balance between performance scores and associated costs.

with the refined prompt. The average costs per test note varied considerably: GPT-4o was the most expensive at $6.70 \times 10^{-2}$, followed by fine-tuning at $4.2 \times 10^{-3}$, Hybrid at $2.2 \times 10^{-3}$, and RAG being the most cost-effective at $1.2 \times 10^{-3}$.

**Impact of RAG Examples:** To validate RAG, we ran an ablation comparing prompt refinement with and without RAG examples. Removing them reduced F1 by 10% for Zephyr and 13% for Phi-3, underscoring their crucial role in boosting performance on clinical notes.

**PCR Analysis:** The right panel in Figure 2 presents the average PCR for refined Phi-3 and Zephyr models across the 12 toxicity symptoms. The updated results reveal a nuanced performance comparison between the two models. For accuracy-based PCR, RAG yielded the highest values, with Zephyr slightly outperforming Phi-3 (494 vs. 484), while the Hybrid approach showed Phi-3 significantly ahead (424 vs. 250). Fine-tuning resulted in higher PCR for Phi-3 (138) compared to Zephyr (98). F1-based PCR values followed a similar pattern, with Zephyr marginally leading in RAG (406 vs. 392) and Phi-3 maintaining a substantial advantage in the Hybrid approach (378 vs. 201). GPT-4, despite high performance, had the lowest PCR due to its higher cost. These results highlight RAG's efficiency in balancing performance and cost for both models, while demonstrating Phi-3's clear advantage in the Hybrid method. The performance gap between models is less pronounced in RAG but more significant in the Hybrid approach, particularly for F1-based metrics, with fine-tuning showing moderate cost-effectiveness for both models.

**Generalizability on Abstract Classification:** To test generalizability, we applied the framework to multi-class medical abstract classification using the Schopf et al. [25] dataset, where each abstract is labeled with one of five disease categories: *neoplasms*, *digestive system diseases*, *nervous system diseases*, *cardiovascular diseases*, or *general pathological conditions*. We trained the models on 10,000 samples with seed 23 and evaluated the final F1-macro scores on the 2,888-sample test set. The results are summarized in Table I. The

TABLE I
F1-MACRO SCORES ON MEDICAL ABSTRACT CLASSIFICATION

| Language Model | Initial | Finetuned | RAG | Hybrid |
|---|---|---|---|---|
| Phi-3 | 0.42 | 0.46 | 0.55 | **0.59** |
| Zephyr-7B | 0.44 | 0.53 | 0.58 | **0.61** |

TABLE II
F1-MACRO SCORES FOR ZEPHYR ACROSS THREE SEEDS

| Method | Seed 23 | Seed 42 | Seed 84 | Mean | Std |
|---|---|---|---|---|---|
| Hybrid | 0.44 | 0.39 | 0.51 | 0.45 | 0.05 |
| Finetuned | 0.34 | 0.33 | 0.42 | 0.36 | 0.04 |
| RAG | 0.48 | 0.51 | 0.39 | 0.46 | 0.05 |

framework produced consistent performance gains for both models across all refinement methods. These positive results on a dataset that differs in domain (abstracts vs. clinical notes) and task complexity provide strong evidence that our refinement methodology can be generalizable.

**Stability Analysis Across Random Seeds:** To rule out random variation, we tested stability by re-running Zephyr with three seeds (23, 42, 84). The final F1 scores for each refinement method are reported in Table II. Final scores are consistent across runs with low variance, confirming that reported gains reflect stable effects of the refinement framework rather than random initialization.

## IV. LIMITATIONS

While our study shows promising results in optimizing compact LLMs for clinical applications, several limitations remain. The iterative refinement with GPT-4 incurs high API costs, suggesting future work could explore more cost-effective, locally-deployable models. Our focus on a basic classification task in clinical notes sets a baseline but leaves room for more complex healthcare applications. Additionally, experiments were limited to prostate cancer radiotherapy; future studies should test generalizability across diverse specialties.

## V. Conclusion

This study investigated iterative refinement strategies for optimizing compact LLMs in clinical applications, with emphasis on privacy, computation, and cost. Using a student–teacher framework with Zephyr and Phi-3 as students and GPT-4o as the teacher, we compared RAG, fine-tuning, and a hybrid approach. RAG consistently improved accuracy and F1-macro while maintaining low cost, and the hybrid method yielded the strongest overall performance, particularly for Phi-3. Notably, Phi-3 often surpassed Zephyr, underscoring that refinement strategy can be more critical than model size. Although GPT-4o achieved the highest performance, its cost limited efficiency. These findings demonstrate that iterative refinement can enable smaller LLMs to achieve competitive performance with substantially greater efficiency, offering a practical alternative to larger models in specialized domains.

## Acknowledgment

This work was supported by the Henry Ford Health + Michigan State University Health Sciences Cancer Seed Funding Program.

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
