# OpenReview forum: "Hybrid Student-Teacher Large Language Model Refinement for Cancer Toxicity Symptom Extraction"
_IEEE.org/EMBS/BHI/2025/Conference — BHI 2025_

### Official Review · Reviewer_7yEt · 2025-06-30
**Enpowering small LLM for clinical data using Student-Teacher LLM refinement**

**Confidence:** 4
**Clarity Of Writing:** great
**Clinical Significance:** great
**Methodological Novelty:** great
**Overall Rating:** 7
**Final Rating:** 7

**Experiments And Results:**

great

**Questions For The Authors:**

Include analysis of the changes in prompts and fine-tuned weights to understand what information most improves student performance.
How does the model handle ambiguous or conflicting notes where symptom status is unclear?
Should this model be online for continuous improvement?
How scalable is this iterative hybrid approach with larger datasets or more diverse symptoms?

**Strengths:**

This study focuses on a real-world challenge in healthcare of privacy, resource constrains, and cost which is highly applicable for clinical adoption. The novel hybrid framework combining prompt refinement with RAG examples and fine-tuning guided by dynamic teacher model. This paper also shows a substantial improvement from their refinement and had a clear cost-benefit analysis including PCR metrics.

**Summary Of The Paper:**

This study addresses the challenge of using large language models for clinical symptom extraction from radiotherapy petient notes while balancing privacy, cost and computational demands. They used student-teacher framework, with GPT-4o as the teachr and smaller LLM (Zephyr-7B-beta and Phi3-mini-128) as students. They investigate interactive refinement strategies including prompt refinement, fine-tuning parameter and a hybrid of both. The experiment is performed on 294 clinical notes of prostate cancer patients and classifing 12 toxicity symptoms after radiotherapy. They found that the Phi3 achieved better F1 and accuracy and that prompt refinement is crucial.

**Weaknesses:**

The current model is limited to classification results which is less ideal in clinical setting. A continuous score of confidence would be more appreciable and clinically useful. Further analysis for how the prompt are refined to improve the performance can be interesting and necessary to be analyzed to make sure the convergence of the changes makes sense.

---

### Official Review · Reviewer_26g1 · 2025-07-02
**Review of “Hybrid Student-Teacher Large Language Model Refinement for Cancer Toxicity Symptom Extraction”**

**Confidence:** 4
**Clarity Of Writing:** great
**Clinical Significance:** great
**Methodological Novelty:** great
**Overall Rating:** 8

**Experiments And Results:**

great

**Questions For The Authors:**

Clarify the specific role of GPT-4o versus the authors in designing and executing the fine-tuning and other refinement steps.

Provide concrete details on the fine-tuning process: dataset size, hyperparameters, number of epochs, validation approach, and runtime cost.

**Strengths:**

The paper addresses a clinically relevant task: structured extraction of toxicity symptoms for oncology documentation and analysis.

The hybrid architecture combining teacher guidance, prompt engineering, fine-tuning, and RAG is conceptually creative.

The use of compact student models (e.g., Phi3) aligns with practical deployment considerations where large LLMs may not be feasible.

The authors demonstrate strong technical command of fine-tuning and model refinement techniques.

**Summary Of The Paper:**

This paper presents a hybrid refinement framework for extracting cancer treatment toxicity symptoms from clinical notes using large language models (LLMs). A teacher model (GPT-4o) reviews outputs from compact student models (Phi3, Zephyr), identifies failure types, and recommends interventions — prompt refinement, fine-tuning, or RAG (retrieval-augmented generation). The authors apply these refinements and report performance improvements, achieving F1-macro up to 0.567 for Phi3 hybrid and 0.442 for Zephyr hybrid.

**Weaknesses:**

The reported gains from fine-tuning are modest (e.g., Phi3 accuracy rising from 0.446 to 0.512; Zephyr from 0.364 to 0.413)

The description of the fine-tuning process lacks concrete details: dataset size, hyperparameters, epochs, validation process, or ablation of its independent contribution.

The role of GPT-4o is somewhat ambiguous — the paper implies it selected treatments, but it seems unlikely GPT-4o autonomously generated training data, configured fine-tuning, or executed the tuning process; this should be clarified.

---

### Official Review · Reviewer_Fawf · 2025-07-09
**Review- Hybrid Student-Teacher Large Language Model Refinement for Cancer Toxicity Symptom Extraction**

**Confidence:** 3
**Clarity Of Writing:** excellent
**Clinical Significance:** good
**Methodological Novelty:** good
**Overall Rating:** 7
**Final Rating:** 7

**Experiments And Results:**

fair

**Questions For The Authors:**

- How does the teacher model (GPT-4o) determine when to switch between prompt refinement and fine-tuning?
Could you elaborate on the criteria or thresholds used to trigger this decision?
- How scalable is your iterative framework for other clinical tasks beyond symptom extraction?
For instance, have you explored applications in diagnosis prediction, temporal event extraction, or medication mapping?
- Given the cost and API dependency of GPT-4o, have you considered using smaller open-source teacher models for local deployment?
What trade-offs did you evaluate in choosing GPT-4o over more accessible alternatives?
- What are the computational and time costs associated with running 16 refinement iterations per epoch?
Could this limit adoption in settings with lower computational resources?

**Strengths:**

This paper presents a practical and efficient method for improving smaller language models in clinical settings using a student-teacher framework. By combining prompt refinement, fine-tuning, and RAG, it enables compact models like Phi-3 and Zephyr to achieve strong performance in extracting cancer toxicity symptoms from clinical notes. The method significantly improves F1 scores while reducing computational costs—Phi-3 is 48× cheaper than GPT-4o. The hybrid refinement approach offers the best balance between accuracy and cost, making it ideal for resource-constrained healthcare environments. Additionally, the framework supports on-premises deployment, addressing privacy concerns common in clinical applications. Overall, the study offers a scalable, cost-effective, and privacy-aware solution for medical NLP tasks.

**Summary Of The Paper:**

The paper introduces a student-teacher framework designed to improve clinical symptom extraction from electronic health records using compact large language models (LLMs). The "teacher" model (GPT-4o) dynamically guides "student" models (Zephyr and Phi-3) through iterative refinement, alternating between prompt engineering, Retrieval-Augmented Generation (RAG), and fine-tuning. Through this process, smaller, cost-effective models achieved significant performance gains—Phi-3 improved F1 scores by 26% and was 48x cheaper than GPT-4o, while Zephyr improved by 13%. Among the techniques tested, RAG was most cost-effective, but the hybrid method (combining prompt refinement and fine-tuning) yielded the best overall performance, especially for Phi-3. This approach addresses challenges in clinical NLP: data privacy, computational cost, and model efficiency—making it suitable for real-world medical applications.

**Weaknesses:**

- The experiments are focused only on prostate cancer radiotherapy notes, which raises concerns about generalizability to other medical conditions, specialties, or institutions.
- The study focuses on a relatively simple symptom classification task (presence, absence, unknown), which doesn’t capture the complexity of many real-world clinical NLP challenges like temporal reasoning, co-reference resolution, or multi-label extraction.
- While the goal is to reduce reliance on large models, the teacher is GPT-4o, which is expensive and requires API access—possibly limiting reproducibility and increasing cost for those trying to adopt the approach at scale.
- The iterative refinement process is computationally intensive, especially with multiple refinement rounds per epoch (up to 16), which could limit real-time or on-demand clinical use—even if the final model is efficient.
- The paper does not test the final models in actual clinical workflows, so the practical impact (e.g., on physician productivity or diagnostic accuracy) remains theoretical.

These limitations were clearly stated by the authors in Section IV.

---

### Official Review · Reviewer_QqmF · 2025-07-12
**Hybrid Student-Teacher Large Language Model Refinement for Cancer Toxicity Symptom Extraction**

**Confidence:** 4
**Clarity Of Writing:** great
**Clinical Significance:** good
**Methodological Novelty:** poor
**Overall Rating:** 4
**Final Rating:** 7

**Experiments And Results:**

good

**Questions For The Authors:**

1. Given that the overall training framework closely follows Khanmohammadi et al. [1], could you clarify what specific methodological innovations your work introduces beyond substituting the student models?

2. What motivated the selection of Zephyr-7b-beta and Phi3-mini-128 as student models? Was there a principled reason for choosing these particular architectures?

3. Have you considered testing your approach on any other datasets or domains to evaluate generalizability? If not, how can we be confident that the improvements are not just dataset-specific?

4. You mention improved cost, but similar methodology were already made in [1]. How this proposed model is more cost effective than previous one? Also, what if the Zephyr-7b-beta and Phi3-mini-128 replaced with some other model (e.g., Gemma-7B, TinyLLaMA, Mistral-7B )? Could you provide the reason  of choosing those two specific models?

5. Can you provide any ablation studies or analysis to attribute the performance gains specifically to your chosen student models, as opposed to random variation or differences in training stability?

6. Could you please clarify whether the 294 clinical notes were derived from unique patients, or whether multiple notes were collected from the same participants over time? If the latter, how was this handled during data splitting to avoid leakage between training and testing?

7. Could you also include computational cost as part of the performance metrics ? This would help contextualize the efficiency claims, especially when comparing different student models.

**Strengths:**

The writing is clear and well-structured, the author explains the methodology effectively, and the presentation of the results is clear.

**Summary Of The Paper:**

The author employed a student-teacher architecture for symptom extraction. The teacher model, GPT-4o, dynamically selects the effective strategy (prompt refinement or fine-tuning) for the student models. The author testedZephyr-7b-beta and Phi3-mini-128 as a student model due to its low cost. Finally, they tested the models performance on 294 clinical notes covering 12 post-radiotherapy toxicity symptoms. The strategy increase F1 score of 26% with 30 times cheaper in cost.

**Weaknesses:**

The main concern with this paper lies in its lack of novelty. The methodology used is nearly identical to that of Khanmohammadi et al. [1]. The authors follow the same teacher–student training framework and apply it to the same dataset without introducing any new task, variation, or significant modification to the original approach. The only major change is the use of different student models—Zephyr-7b-beta and Phi3-mini-128—instead of Mixtral-8x7B. While this results in improved performance, simply swapping model architectures does not, on its own, represent a novel contribution.

The authors also claim that their approach reduces computational cost, but this was already addressed in Khanmohammadi et al. [1]. Without further analysis or evidence to demonstrate a new or more efficient training procedure, this claim feels redundant.

Overall, the contribution of the paper is limited. While the results are slightly better, the lack of new methodology, new data, or meaningful insight makes the work feel more like an incremental update than a novel research contribution.

reference:
[1]. R. Khanmohammadi, A. I. Ghanem, K. Verdecchia, R. Hall, M. Elshaikh, B. Movsas, H. Bagher-Ebadian, I. Chetty, M. M. Ghassemi, and K. Thind, “Iterative prompt refinement for radiation oncology symptom extraction using teacher-student large language models,” 2024. [Online]. Available: https://arxiv.org/abs/2402.04075